# GENERATIVE PREDECESSOR MODELS FOR SAMPLE-EFFICIENT IMITATION LEARNING

**Yannick Schroecker***
College of Computing
Georgia Institute of Technology
Atlanta, USA
yannickschroecker@gatech.edu

**Mel Vecerik & Jonathan Scholz**
DeepMind
London, United Kingdom
{vec,jscholz}@google.com

## ABSTRACT

We propose Generative Predecessor Models for Imitation Learning (GPRIL), a novel imitation learning algorithm that matches the state-action distribution to the distribution observed in expert demonstrations, using generative models to reason probabilistically about alternative histories of demonstrated states. We show that this approach allows an agent to learn robust policies using only a small number of expert demonstrations and self-supervised interactions with the environment. We derive this approach from first principles and compare it empirically to a state-of-the-art imitation learning method, showing that it outperforms or matches its performance on two simulated robot manipulation tasks and demonstrate significantly higher sample efficiency by applying the algorithm on a real robot.

## 1 INTRODUCTION

Training or programming agents to act intelligently in unstructured and sequential environments is a difficult and central challenge in the field of artificial intelligence. Imitation learning provides an avenue to tackle this challenge by allowing agents to learn from human teachers, which constitutes a natural way for experts to describe the desired behavior and provides an efficient learning signal for the agent. It is thus no surprise that imitation learning has enabled great successes on robotic (Chernova and Thomaz, 2014) as well as software domains (e.g. Aytar et al. (2018)). Yet, key challenges in the field are diverse and include questions such as how to learn from observations alone (e.g. Aytar et al. (2018)), learning the correspondence between the expert's demonstrations and the agent's observations (e.g. Sermanet et al. (2017)) as well as the question of how to integrate imitation learning with other approaches such as reinforcement learning (e.g. Vecerik et al. (2017)). However, at the core of the imitation learning problems lies the challenge of utilizing a given set of demonstrations to match the expert's behavior as closely as possible. In this paper, we approach this problem considering the setting where the set of expert demonstrations is given up-front and the dynamics of the environment can only be observed through interaction.

In principle, imitation learning could be seen as a supervised learning problem, where the demonstrations are used to learn a mapping from observed states to actions. This solution approach is known as behavioral cloning. However, it has long been known that the sequential structure of the task admits more effective solutions. In particular, the assumptions made in supervised learning are restrictive and don't allow the agent to reason about the effect of its actions on it's future inputs. As a result, errors and deviations from demonstrated behavior tend to accumulate over time as small mistakes lead the agent to parts of the observation space that the expert has not explored (Ross and Bagnell, 2010). In this work, we propose a novel imitation learning algorithm, Generative Predecessor Models for Imitation Learning (GPRIL), based on a simple core insight: Augmenting the training set with state-action pairs that are likely to eventually lead the agent to states demonstrated by the expert is an effective way to train corrective behavior and to prevent accumulating errors.

Recent advances in generative modeling, such as Goodfellow et al. (2014); Kingma and Welling (2013); Van Den Oord et al. (2016b;a); Dinh et al. (2016), have shown great promise at modeling complex distributions and can be used to reason probabilistically about such state-action pairs.

---

*This work was carried out at DeepMind.

Specifically, we propose to utilize Masked Autoregressive Flows (Papamakarios et al., 2017) to model long-term predecessor distributions, i.e. distributions over state-action pairs which are conditioned on a state that the agent will see in the future. Predecessor models have a long history in reinforcement learning (e.g. Peng and Williams (1993)) with recent approaches using deep networks to generate off-policy transitions (Edwards et al., 2018; Pan et al., 2018) or to reinforce behavior leading to high-value states (Goyal et al., 2018). Here, we use predecessor models to derive a principled approach to state-distribution matching and propose the following imitation learning loop:

1. Interact with the environment and observe state, action as well as a future state. To encode long-term corrective behavior, these states should be multiple steps apart.

2. Train a conditional generative model to produce samples like the observed state-action pair when conditioned on the observed future state.

3. Train the agent in a supervised way, augmenting the training set using data drawn from the model conditioned on demonstrated states. The additional training data shows the agent how to reach demonstrated states, enabling it to recover after deviating from expert behavior.

In the above, we laid out the sketch of an algorithm that intuitively learns to reason about the states it will observe in the future. In section 3, we derive this algorithm from first principles as a maximum likelihood approach to matching the state-action distribution of the agent to the expert's distribution. In section 4, we compare our approach to a state-of-the-art imitation learning method (Ho and Ermon, 2016) and show that it matches or outperforms this baseline on our domains while being significantly more sample efficient. Furthermore, we show that GPRIL can learn using demonstrated states alone, allowing for a wider variety of methods to be used to record demonstrations. Together these properties are sufficient to allow GPRIL to be applied in real-world settings, which we demonstrate in section 4.3. To our knowledge this is the first instance of dynamic, contact-rich and adaptive behavior being taught solely using the kinesthetic-teaching interface of a collaborative robot, without resorting to tele-operation, auxiliary reward signals, or manual task-decomposition.

## 2 BACKGROUND

### 2.1 MARKOV DECISION PROCESSES WITHOUT REWARDS

As is usual, we model the problem as a Markov decision process without reward. That is, given state and action sets $\mathcal{S}, \mathcal{A}$, the agent is observing states $s \in \mathcal{S}$ and taking actions $a \in \mathcal{A}$. In this work, we use $s$ and $a$ to refer to states and actions observed during self-supervision and $\bar{s} \in \mathcal{S}$ and $\bar{a} \in \mathcal{A}$ to refer to target and demonstration states and actions. We furthermore use superscripts $\bar{s}^{(i)}, \bar{a}^{(i)}$ to refer to specific instances, e.g. specific demonstrated state-action pairs, and subscripts, e.g. $s_t, a_t$, to indicate temporal sequences. The observed transitions are guided by the Markovian dynamics of the environment and the probability of transitioning from state $s$ to state $s'$ by taking action $a$ is denoted as $p(s_{t+1} = s'|s_t = s, a_t = a)$. The agent's behavior is defined by a stationary parametric policy $\pi_\theta(a|s)$ while the expert's behavior is modeled by a stationary distribution $\pi^*(\bar{a}|\bar{s})$. We denote as $d_t^\pi(s)$ the probability of observing state $s$ at time-step $t$ when following policy $\pi$. Under the usual ergodicity assumptions, each such policy induces a unique stationary distribution of observed states $d^\pi(s) = \lim_{t\to\infty} d_t^\pi(s)$ as well as a stationary joint state-action distribution $\rho^\pi(s,a) := \pi(a|s)d^\pi(s)$. Furthermore, we use $q_t^\pi$ to refer to the dynamics of the time reversed Markov chain induced by a particular policy $\pi$ at time-step t

$$q_t^\pi(s_t = s, a_t = a|s_{t+1} = s') = d_{t+1}(s')^{-1}d_t(s)\pi(a_t|s_t)p(s_{t+1} = s'|s_t = s, a_t = a) \quad (1)$$

and define $q^\pi(s_t = s, a_t = a|s_{t+1} = s') := \lim_{t\to\infty} q_t^\pi(s_t = s, a_t = a|s_{t+1} = s')$. For the purposes of this work, we handle the episodic case with clear termination conditions by adding artificial transitions from terminal states to initial states. This creates a modified, ergodic MDP with identical state-distribution and allows us to assume arbitrarily large $t$ such that $q_t^\pi = q^\pi$. Finally, we extend this notation to multi-step transitions by writing $q^\pi(s_t = s, a_t = a|s_{t+j} = s')$.

### 2.2 IMITATION LEARNING

In this work, we are considering two settings of imitation learning. In the first setting, the agent is given a set of observed states $\bar{s}^{(1)}, \bar{s}^{(2)}, \cdots, \bar{s}^{(N)}$ and observed corresponding actions

$\bar{a}^{(1)}, \bar{a}^{(2)}, \cdots, \bar{a}^{(N)}$ as expert demonstrations. The goal in this setting is to learn a policy $\pi_\theta$ that matches the expert's behavior as closely as possible. In the second setting, the agent is given the states observed by the expert but is not aware of the actions the expert has taken. Recent years have seen heightened interest in a related setting where to goal is to track expert state trajectories (Zhu et al., 2018; Peng et al., 2018; Pathak et al., 2018). These approaches do not learn general policies that can adapt to unseen situations. A straightforward approach to train general policies in the first setting, usually referred to as behavioral cloning, is to treat the task as a supervised learning problem (e.g. Pomerleau (1989)). However, as outlined in section 1, predictions made by $\pi_\theta(a|s)$ influence future observations thus violating a key assumption of supervised learning, which states that inputs are drawn from an i.i.d. distribution. This has formally been analyzed by Ross et al. who introduce a family of algorithms (e.g. Ross and Bagnell (2010); Ross et al. (2011)) that provably avoid this issue. However, these approaches require the expert to continuously provide demonstrations and thus are not applicable when the set of demonstrations is assumed to be fixed. A popular avenue of research that considers this setting is inverse reinforcement learning (IRL). Inverse reinforcement learning (Ng and Russell, 2000) aims to learn a reward function for which $\pi^*$ is optimal and thus captures the intent of the expert. The arguably most successful approach to IRL aims to match the state-action distribution to that of the demonstrations (Ziebart et al., 2008) with recent approaches extending these ideas to the model free case (Boularias et al., 2011; Finn et al., 2016; Fu et al., 2018).

However, inverse reinforcement learning is indirect and ill-defined as many reward functions induce the same behavior. Recently, methods have been proposed that aim to match state-action distributions directly and achieve state-of-the-art result without learning a reward function first. Generative Adversarial Imitation Learning (GAIL) (Ho and Ermon, 2016) uses an adversarial objective, training a discriminator to identify demonstrations and using TRPO to train a policy that fools the discriminator. While GAIL is able to achieve impressive results, the adversarial objective can make the learning procedure unstable and unpredictable. This is especially true when parts of the state-space are not under the agent's control, yielding a setting resembling a conditional GAN (Mirza and Osindero, 2014), which are prone to issues such mode collapse (Odena et al., 2017). We compare our approach with GAIL in section 4. State Aware Imitation Learning (SAIL) (Schroecker and Isbell, 2017) is an alternative method that aims to learn the gradient of the state-action distribution using a temporal-difference update rule. This approach is able to avoid instabilities prompted by the adversarial learning rule but is only applicable to policies with a small number of parameters where learning a representation of the gradient is feasible. In this work, we follow a gradient descent approach similar to SAIL but estimate the gradient without representing it explicitly by a neural network. These methods can also be used to match state-distributions without actions. While the order of states is unspecified and the induced policy is therefore not unique, including transition information such as velocities can allow the agent to learn solely from expert state trajectories nonetheless.

## 2.3 GENERATIVE MODELS

Recent years have seen great advances in deep generative models. A variety of approaches such as Generative Adversarial Networks (Goodfellow et al., 2014), Variational Auto Encoders (Kingma and Welling, 2013), autoregressive networks (e.g. Germain et al. (2015); Van Den Oord et al. (2016b;a) and normalizing flows (Dinh et al., 2016) have been proposed which enable us to learn complex distributions and efficiently generate samples. In this work, we use generative models to model the distribution of long-term predecessor state-action pairs. While the approach we propose is model agnostic, we choose to model this distribution using masked autoregressive flows (Papamakarios et al., 2017) (MAF). MAFs are trained using a maximum likelihood objective, which allows for a stable and straight-forward training procedure. Autoregressive models are capable of representing complex distributions $p(x); x \in \mathbb{R}^n$ by factoring the distribution $p(x) = p_1(x_1) \prod_{i=1}^{N-1} p_{i+1}(x_{i+1}|x_1, \ldots, x_i)$ and learning a model for each $p_i$. In this paper, we model each $x_i$ to be distributed by $x_i \sim \mathcal{N}(\cdot|\mu_i, \sigma_i)$ where each $\mu_i$ and $\sigma_i$ is a function of $x_{1:i-1}$. Masked autoencoders (Germain et al., 2015) provide a straight-forward approach to parameter sharing and allow representing these functions using a single network. MAFs stack multiple autoregressive models with different orderings and thus avoid the strong inductive bias imposed by the order of variables. Using the reparameterization trick, the autoregressive model can be seen as a deterministic and invertible transformation of a random variable: $x = f(z) := \mu + \sigma z; z \sim \mathcal{N}(\cdot|0, 1)$. The change of variable formula then allows us to calculate the density of $x$:

$$\log p(x) = \log p_{\mathcal{N}}(f^{-1}(x)) + \log \det(|J(f^{-1}(x))|) \tag{2}$$

---

**Algorithm 1** Generative Predecessor Models for Imitation Learning (GPRIL)

---

1: **function** GPRIL($N_\mathcal{B}$, $N_\pi$, $B$ (*batch size*))
2:      **for** $i \leftarrow 0..\#$Iterations **do**
3:          **for** $k \leftarrow 0..N_\mathcal{B}$ **do**
4:              **for** $n \leftarrow 0..B$ **do**
5:                  Sample $s_t^{(n)}, a_t^{(n)}$ from replay buffer
6:                  Sample $j \sim Geom(1 - \gamma)$
7:                  Sample $s_{t+j}^{(n)}$ from replay buffer
8:              Update $\omega_s$ using gradient $\sum_{n=0}^{B} \nabla_{\omega_s} \log \mathcal{B}_{\omega_s}^s(s_t^{(n)}|s_{t+j}^{(n)})$
9:              Update $\omega_a$ using gradient $\sum_{n=0}^{B} \nabla_{\omega_a} \log \mathcal{B}_{\omega_a}^a(a_t^{(n)}|s_t^{(n)}, s_{t+j}^{(n)})$
10:          **for** $k \leftarrow 0..N_\pi$ **do**
11:              **for** $n \leftarrow 0..B$ **do**
12:                  Sample $\overline{s}^{(n)}, \overline{a}^{(n)}$ from expert demonstrations
13:                  Sample $s^{(n)} \sim \mathcal{B}_{\omega_s}^s(\cdot|\overline{s}^{(n)})$, $a^{(n)} \sim \mathcal{B}_{\omega_a}^a(\cdot|s^{(n)}, \overline{s}^{(n)})$
14:              Update $\theta$ using gradient $\sum_{n=0}^{B} \beta_\pi \nabla_\theta \log \pi_\theta(\overline{a}^{(n)}|\overline{s}^{(n)}) + \beta_d \nabla_\theta \log \pi_\theta(a^{(n)}|s^{(n)})$

---

Where the autoregressive nature of $f$ in eq. 2 allows for tractable computation of the second term. MAFs chain multiple such transformations to derive highly expressive explicit density models able to model complex dynamics between target states and long-term predecessor state-action pairs.

## 3 GPRIL

In section 1, we provided an intuitive framework for using predecessor models to augment our training set and achieve robust imitation learning from few samples. In this section, we will derive this algorithm based on state-action distribution matching. To this end, we first derive the gradient of the logarithmic state distribution based on samples from a long-term predecessor distribution that we will define below. In section 3.2, we describe how to train a generative model of the predecessor distribution, which will allow us to evaluate this gradient. Ascending on this gradient evaluated at demonstrated states leads the agent to stick to those states and thus provides a corrective measure (Schroecker and Isbell, 2017). Furthermore, reproducing the states of the expert can be sufficient to achieve the correct behavior if the state-space is chosen appropriately as we will show in section 4. We will show how to use this gradient to match state-action-distributions in section 3.3.

### 3.1 ESTIMATING THE GRADIENT OF THE STATE DISTRIBUTION

Here, we will show that the samples drawn from a long-term predecessor distribution conditioned on $\overline{s}$ enable us to estimate the gradient of the logarithmic state distribution $\nabla_\theta \log d^{\pi_\theta}(\overline{s})$ and, later, to match the agent's state-action-distribution to that of the expert. To achieve this goal, we can utilize the fact that the stationary state distribution of a policy can be defined recursively in terms of the state distribution at the previous time step, similar to Morimura et al. (2010):

$$d^{\pi_\theta}(\overline{s}) = \int d^{\pi_\theta}(s)\pi_\theta(a|s)p(s_{t+1} = \overline{s}|s_t = s, a_t = a)ds, a. \tag{3}$$

Taking the derivative shows that this notion extends to the gradient as well as its logarithm:

$$\nabla_\theta d^{\pi_\theta}(\overline{s}) = \int \rho^{\pi_\theta}(s, a)p(s_{t+1} = \overline{s}|s_t = s, a_t = a)\left(\nabla_\theta \log d^{\pi_\theta}(s) + \nabla_\theta \log \pi_\theta(a|s)\right)ds, a \tag{4}$$

$$\nabla_\theta \log d^{\pi_\theta}(\overline{s}) = \int q^{\pi_\theta}(s_t = s, a_t = a|s_{t+1} = \overline{s})\left(\nabla_\theta \log d^{\pi_\theta}(s) + \nabla_\theta \log \pi_\theta(a|s)\right)ds, a \tag{5}$$

The recursive nature of this gradient then allows us to unroll the gradient indefinitely. However, this process is cumbersome and will be left for appendix A. We arrive at the following equality:

$$\nabla_\theta \log d^{\pi_\theta}(\overline{s}) = \lim_{T \to \infty} \int \sum_{j=0}^{T} q^{\pi_\theta}(s_t = s, a_t = a|s_{t+j+1} = \overline{s})\nabla_\theta \log \pi_\theta(a|s)ds, a \tag{6}$$

The derivation of our approach now rests on two key insights: First, in ergodic Markov chains, such as the ones considered in our setting, decisions that are made at time $t$ affect the probability of seeing

state $\overline{s}$ at time $t+j$ more strongly if $j$ is small. In the limit, as $j \to \infty$, the expectation of the gradient $\nabla_\theta \log \pi_\theta(a_t|s_t)$ vanishes and the decision at time $t$ only adds variance to the gradient estimate. Introducing a discount factor $\gamma$ similar to common practice in reinforcement learning (Sutton and Barto, 1998) places more emphasis on decisions that are closer in time and can thus greatly reduce variance. We explore this interpretation further in appendix C. Second, by introducing a discount factor, the effective time-horizon is now finite. This allows us to replace the sum over all states and actions in each trajectory with a scaled expectation over state-action pairs. Formally, we can write this as follows and arrive at our main result:

$$\nabla_\theta \log d^{\pi_\theta}(\overline{s}) \approx \int \sum_{j=0}^{\infty} \gamma^j q^{\pi_\theta}(s_t = s, a_t = a|s_{t+j+1} = \overline{s}) \nabla_\theta \log \pi_\theta(a|s) ds, a \tag{7}$$

$$\propto \mathbb{E}_{s,a \sim \mathcal{B}^{\pi_\theta}(\cdot,\cdot|\overline{s})} \left[ \nabla_\theta \log \pi_\theta(a|s) \right]$$

where $\mathcal{B}^{\pi_\theta}$ corresponds to the long-term predecessor distribution modeling the distribution of states and actions that, under the current policy $\pi_\theta$, will eventually lead to the given target state $\overline{s}$:

$$\mathcal{B}^{\pi_\theta}(s,a|\overline{s}) := (1-\gamma) \sum_{j=0}^{\infty} \gamma^j q^{\pi_\theta}(s_t = s, a_t = a|s_{t+j+1} = \overline{s}) \tag{8}$$

### 3.2 Long-term generative predecessor models

In the previous section, we derived the gradient of the logarithm of the stationary state distribution as approximately proportional to the expected gradient of the log policy, evaluated at samples obtained from the long-term predecessor distribution $\mathcal{B}^{\pi_\theta}$. In this work, we propose to train a model $\mathcal{B}^{\pi_\theta}_\omega$ to represent $\mathcal{B}^{\pi_\theta}$ and use its samples to estimate $\nabla_\theta \log d^{\pi_\theta}(\overline{s})$. However, rather than unrolling a time-reversed Markov model in time, which is prone to accumulated errors, we propose to use a generative model to directly generate jumpy predictions. We have furthermore found that imposing a sensible order on autoregressive models achieves good results and thus propose to use two conditional MAFs (Papamakarios et al., 2017) $\mathcal{B}^s_{\omega_s}, \mathcal{B}^a_{\omega_a}$ in a factored representation:

$$\mathcal{B}^{\pi_\theta}_\omega(s,a|\overline{s}) := \mathcal{B}^s_{\omega_s}(s|\overline{s})\mathcal{B}^a_{\omega_a}(a|s,\overline{s}). \tag{9}$$

To train this model, we collect training data using self-supervised roll-outs: We sample states, actions and target-states where the separation in time between the state and target-state is selected randomly based on the geometric distribution parameterized by $\gamma$ as a training set for $\mathcal{B}^{\pi_\theta}_\omega$.

Training data for $\mathcal{B}^s_{\omega_s}, \mathcal{B}^a_{\omega_a}$ are obtained by executing the current policy to obtain a sequence $s_0, a_0, s_1, a_1, \cdots$, which we store in a replay buffer. In practice, we store data from multiple iterations in this buffer in order to decrease the variance of the gradient. While our algorithm does not explicitly account for off-policy samples, we found empirically that a short replay buffer does not degrade final performance while significantly improving sample efficiency. To obtain a training sample, we first pick $s = s_t$ and $a = a_t$ for a random $t$. We now select a future state $\overline{s} = s_{t+j+1}$ from that sequence. For any particular $s_{t+j+1}$ we now have $s, a \sim q^{\pi_\theta}_t(s_t = \cdot, a_t = \cdot|s_{t+j+1} = \overline{s}) \approx q^{\pi_\theta}(s_t = \cdot, a_t = \cdot|s_{t+j+1} = \overline{s})$. Note that in the episodic case, we can add transitions from terminal to initial states and pick $t$ to be arbitrarily large such that the approximate equality becomes exact (as outlined in section 2.1). In non-episodic domains, we find the approximation error to be small for most $t$. Finally, we choose $j$ at random according to a geometric distribution $j \sim Geom(1-\gamma)$ and have a training triple $s, a, \overline{s}$ that can be used to train $\mathcal{B}^a_{\omega_a}$ and $\mathcal{B}^s_{\omega_s}$ as it obeys

$$s, a \sim (1-\gamma) \sum_{j=0}^{\infty} \gamma^j q^{\pi_\theta}_t(s_t = \cdot, a_t = \cdot|s_{t+j+1} = \overline{s}) = \mathcal{B}^{\pi_\theta}(\cdot,\cdot|\overline{s}). \tag{10}$$

### 3.3 Matching state-action distributions with GPRIL

State-action distribution matching has been a promising approach to sample-efficient and robust imitation learning (see section 2.2). While each policy induces a unique distribution of states and behavioral cloning would therefore be sufficient in the limit, it is sub-optimal the case of limited data. Matching the joint-distribution directly ensures that we minimize discrepancies between everything we observed from the expert and the behavior the agent exhibits. In this work, we propose

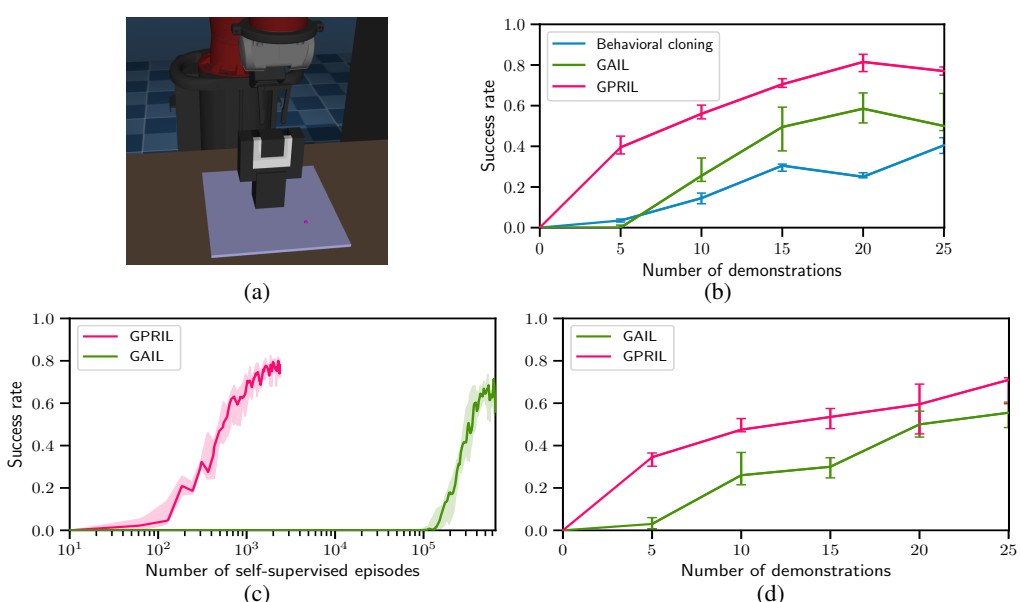

Figure 1: **a)** Depiction of the clip-insertion task. **b)** Median final success rate and interquartile range out of 100 roll-outs over 8 seeds. GPRIL achieves the highest success rate followed by GAIL. **c)** Median final success rate and IQR on clip insertion comparing sample efficiency. GPRIL is able to solve the task using several orders of magnitude fewer environment interactions. **d)** Comparison on clip insertion trained on states alone. Learning from states alone only slightly affects performance.

a maximum likelihood based approach, ascending on the estimated gradient of the joint distribution:

$$\nabla_\theta \log \rho^{\pi_\theta}(\overline{s}, \overline{a}) = \nabla_\theta \log \pi_\theta(\overline{a}|\overline{s}) + \nabla_\theta \log d^{\pi_\theta}(\overline{s}). \tag{11}$$

where $\nabla_\theta \log \pi_\theta(\overline{a}|\overline{s})$ can be computed directly by taking the gradient of the policy using the demonstrated state-action pairs and $\nabla_\theta \log d^{\pi_\theta}(\overline{s})$ can be evaluated using samples drawn from $\mathcal{B}_\omega^{\pi_\theta}(\cdot, \cdot|\overline{s})$ according to equation 7. We introduce scaling factors $\beta_\pi$ and $\beta_d$ to allow for finer control, interpolating between matching states only ($\beta_\pi = 0$) and behavioral cloning ($\beta_d = 0$) and have:

$$\nabla_\theta \log \rho^{\pi_\theta}(\overline{s}, \overline{a}) \approx \beta_\pi \log \pi_\theta(\overline{a}|\overline{s}) + \beta_d \mathbb{E}_{s,a \sim \mathcal{B}^{\pi_\theta}(\cdot, \cdot|\overline{s})} \left[ \nabla_\theta \log \pi_\theta(a|s) \right]. \tag{12}$$

Here, higher values of $\beta_\pi$ provide more supervised guidance while lower values aim to prevent accumulating errors. This gives rise to the full-algorithm: We fill the replay buffer by asynchronously collecting experience using the current policy. Simultaneously, we repeatedly draw samples from the replay buffer to update the predecessor models and use expert samples in combination with an equal number of artificial samples to update the policy. This procedure is described fully in algorithm 1.

## 4 EXPERIMENTS

To evaluate our approach, we use a range of robotic insertion tasks similar to the domains introduced by Vecerik et al. (2017) but without access to a reward signal or, in some cases, expert actions. We choose these domains both for their practical use, and because they highlight challenges faced when applying imitation learning to the real world. Specifically, collecting experience using a robot arm is costly and demands efficient use of both demonstrations and autonomously gathered data. Furthermore, insertion tasks typically require complex searching behavior and cannot be solved by open-loop tracking of a given demonstration trajectory when the socket position is variable. We first compare against state-of-the-art imitation learning methods on a simulated clip insertion task, then explore the case of extremely sparse demonstrations on a simulated peg insertion task and finally, demonstrate real-world applicability on its physical counterpart.

### 4.1 CLIP INSERTION

In the first task, a simulated robot arm has to insert an elastic clip into a plug, which requires the robot to first flex the clip in order to be able to insert it (see figure 1a). In real-world insertion tasks, the

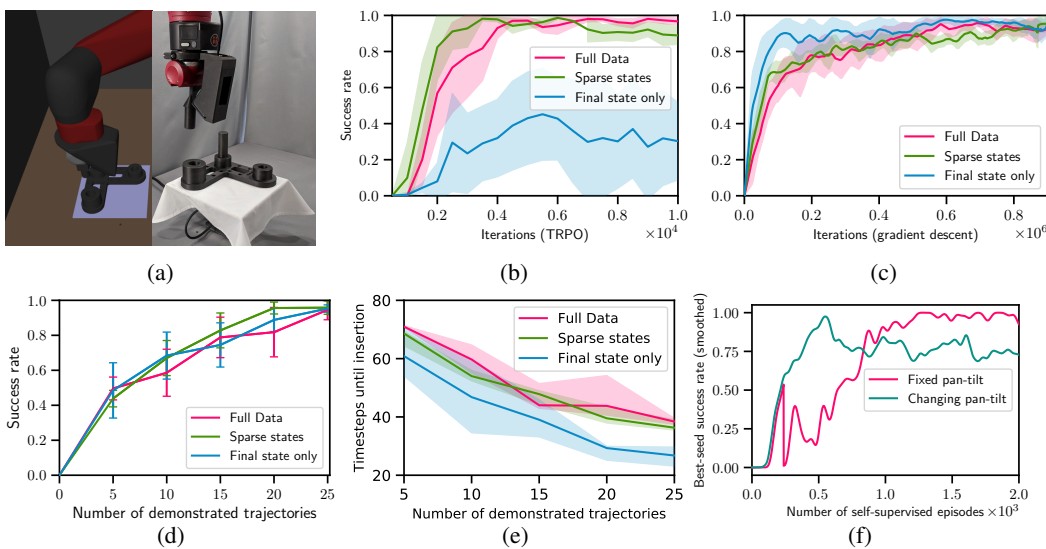

Figure 2: **a)** Depiction of the peg insertion task **b)** Average success rate and 95% confidence interval of GAIL with 25 demonstrations across 10 runs (evaluated over 100 roll-outs). **c)** Average success rate of GPRIL across 5 seeds. Unlike GAIL, the performance of GPRIL doesn't drop off when provided with only final states. **d)** Average success rate and confidence interval of GPRIL. Final performance after $10^6$ iterations increases steadily as the number of demonstrated trajectory increases but is unaffected by dropping steps from each demonstration. **e)** Median length and IQR of trajectories that are successfully inserting the peg. Providing only final states is significantly faster. **f)** Best seed performance on both variations of peg insertion on the real robot.

pose of the robot, the socket, or the grasped object may vary. We capture this variability by mounting the socket on a pan-tilt unit, which is randomized by $\pm 0.8$ (pan) and $\pm 0.2$ radians (tilt). To perform this behavior, the robot observes proprioceptive features, specifically joint position, velocity and torques as well as the position of the end-effector and the socket orientation as a unit quaternion. The task terminates when the robot leaves the work-space, reaches the goal, or after 50 seconds.

For comparative evaluation, we train a policy network to predict mean and variance, modelling a multivariate normal distribution over target velocities and train it using GPRIL, GAIL as well as behavioral cloning. We record expert demonstrations using tele-operation and normalize observations based on the recorded demonstrations. We then train GPRIL using a single asynchronous simulation and compare against the open source implementation of GAIL[1] for which we use 16 parallel simulations. We select the best hyper parameters for GAIL found on a grid around the hyperparameters used by Ho and Ermon (2016) but lower the batch size to 256 as it increases the learning speed and accounts for the significantly slower simulation of the task. We furthermore enable bootstrapping regardless of whether or not the episode terminated. As all discriminator rewards are positive, handling terminal transitions explicitly can induce a bias towards longer episodes. This is beneficial in the domains used by Ho and Ermon but harmful in domains such as ours where the task terminates on success. A detailed list of hyper-parameters can be found in appendix B.

We report final results after convergence and can see in figure 1b that both GAIL and GPRIL outperform behavioral cloning, indicating that generalizing over state-action trajectories requires fewer demonstrations than generalizing over actions alone. Furthermore, we observe a higher success rate using GPRIL and find that policies trained using GPRIL are more likely to retry insertion if the robot slides the clip past the insertion point. To compare sample efficiency of GPRIL to GAIL, we limit the rate at which the asynchronous actor is collecting data. While sample efficiency of GAIL could be increased by decreasing batch size or increasing various learning rates, we found that this can lead to unstable learning performance while reducing the amount of samples required by only a small amount. As can be seen in figure 1c, GPRIL requires several orders of magnitudes fewer environment interactions to learn this task. Finally, we evaluate the case where the expert's actions

---

[1]https://github.com/openai/baselines/tree/master/baselines/gail

are unknown. Since the state-space includes information about joint velocities as well as positions, we find that matching the state-distribution is sufficient to solve the task. GPRIL can achieve this by setting $\beta_d = 1$ and $\beta_\pi = 0$. As can be seen in figure 1d, performance deteriorates only marginally with a similar difference in performance between both methods.

## 4.2 PEG INSERTION WITH PARTIAL DEMONSTRATIONS

The second task is a simulated version of the peg-insertion task depicted in figure 2a. In this task, the robot has to insert the peg into the hole, which is again mounted on a pan-tilt that randomly assumes pan and tilt angles varying by 0.4 and 0.1 respectively. Hyperparameters are largely identical and we report minor differences in appendix B. Observation and action space are identical with the exception of the omission of torques from the observation space as they are not necessary to solve this task. We use this task to evaluate the performance of GAIL and GPRIL when learning from only a very limited set of demonstrated states. To this end, we compare three different scenarios in which the demonstrations are sparsified to varying degrees: In the first case, the agent has access to the full state-trajectories of the expert, in the second only every tenth state is available and in the third the agent sees only the final state of each of the 25 trajectories. Being able to learn from only partial demonstrations is a useful benchmark for the effectiveness of imitation learning methods but can also provide a convenient way of providing demonstrations and can free the agent to find more optimal trajectories between states (see for example Akgun et al. (2012); Schroecker et al. (2016)). As can be seen in figures 2d and 2e, GPRIL achieves similar final success rates in all three scenarios while being able to learn a significantly faster insertion policy when learning from final states alone. We find that in the first two scenarios, this holds for GAIL as well as can been in figure 2b while in the third case, GAIL becomes highly unstable and the resulting performance can vary wildly, leading to a low average success rate. We hypothesize that these instabilities are a result of the discriminator overfitting to the very small amount of negative samples in its training data.

## 4.3 PEG INSERTION ON A PHYSICAL SYSTEM

In previous sections we demonstrated sample-efficiency that indicates applicability of GPRIL to real-world physical systems. To test this, we evaluate our approach on two variations of the physical peg-insertion task depicted in figure 2a, involving a physical pan-tilt unit which is fixed in one scenario and has pan and tilt angles varying by 0.1 and 0.02 radians in the second scenario. For each scenario, we provide 20 demonstrations using kinesthetic teaching, which constitutes a natural way of recording demonstrations but provides state-trajectories only (Chernova and Thomaz, 2014). Hyper-parameters are altered from section 4.2 to trade off a small margin of accuracy for higher learning speeds and are reported in appendix B. Note, however, that tuning hyper-parameters precisely is very difficult on a physical system. As can be seen in figure 2f, GPRIL is able to learn a successful insertion policy that generalizes to unseen insertion angles using just a few hours of environment interactions[2]. We report best-seed performance as we observe a high amount of variability due to factors outside the agent's control, such as the pan-tilt unit not reporting accurate information after physical contact with the robot. However, we wish to point out that the increased difficulty due to less predictable control is also likely to introduce additional variance that could be reduced further with careful design of exploration noise and other hyper-parameters. We furthermore provide a video of the training procedure and final policy to highlight the efficiency of our method[3].

## 5 CONCLUSION

We introduced GPRIL, a novel algorithm for imitation learning which uses generative models to model multi-step predecessor distributions and to perform state-action distribution matching. We show that the algorithm compares favorably with state-of-the-art imitation learning methods, achieving higher or equivalent performance while requiring several orders of magnitude fewer environment samples. Importantly, stability and sample-efficiency of GPRIL are sufficient to enable experiments on a real robot, which we demonstrated on a peg-insertion task with a variable-position socket.

---

[2]Total time to collect and train on 2000 roll-outs was 18.5 and 16.5 hours on the fixed and changing versions of the task respectively. However, GPRIL converged to good policies significantly sooner.

[3]https://youtu.be/Dm0OCNujEmE

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

## A   DERIVATION OF EQUATION 6

Here, we derive equation 6 which unrolls the recursive definition of $\nabla_\theta \log d^{\pi_\theta}(\bar{s})$ and rewrites it such that it can be replaced by an expectation over states and actions along trajectories leading to the state $\bar{s}$. In section 3.1, we derive a recursive definition of $\nabla_\theta \log d^{\pi_\theta}(\bar{s})$ which we will restate in more detail:

$$\nabla_\theta d^{\pi_\theta}(\bar{s}) = \int \nabla_\theta \rho^{\pi_\theta}(s) p(s_{t+1} = \bar{s}|s_t = s, a_t = a) ds, a$$

$$d^{\pi_\theta}(\bar{s}) \nabla_\theta \log d^{\pi_\theta}(\bar{s}) = \int \rho^{\pi_\theta}(s, a) p(s_{t+1} = \bar{s}|s_t = s, a_t = a) \nabla_\theta \log \rho^{\pi_\theta}(s, a) ds, a$$

$$\nabla_\theta \log d^{\pi_\theta}(\bar{s}) = \int q^{\pi_\theta}(s_t = s, a_t = a|s_{t+1} = \bar{s}) \left( \nabla_\theta \log d^{\pi_\theta}(s) + \nabla_\theta \log \pi_\theta(a|s) \right) ds, a$$

We can now unroll this definition:

$$\nabla_\theta \log d^{\pi_\theta}(\bar{s}) = \int q^{\pi_\theta}(s_t = s, a_t = a|s_{t+1} = \bar{s}) \left( \nabla_\theta \log d^{\pi_\theta}(s) + \nabla_\theta \log \pi_\theta(a|s) \right) ds, a$$

$$
\begin{aligned}
= \lim_{T \to \infty} \int \Bigg( & q^{\pi_\theta}(s_{t+T-1}, a_{t+T-1}|s_{t+T} = \bar{s}) \prod_{j=0}^{T-2} q^{\pi_\theta}(s_{t+j}, a_{t+j}|s_{t+j+1}) \\
& \sum_{j=0}^{T} \nabla_\theta \log \pi_\theta(a_{t+j}|s_{t+j}) \Bigg) ds_{t:t+T-1}, a_{t:t+T-1} + \\
& \int q^{\pi_\theta}(s_t = s, a_t = a|s_{t+T} = \bar{s}) \nabla_\theta \log d^{\pi_\theta}(s) ds, a
\end{aligned}
\tag{13}
$$

Note that $\lim_{T \to \infty} q^{\pi_\theta}(s_t = s, a_t = a|s_{t+T} = \bar{s}) = \rho^{\pi_\theta}(s, a)$ due to Markov chain mixing and, therefore, the second term of the above sum reduces to 0 as

$$\int d^{\pi_\theta}(s) \pi(a|s) \nabla_\theta \log d^{\pi_\theta}(s) ds, a = 0. \tag{14}$$

By pulling out the sum, we can now marginalize out most variables and shift indices to arrive at the desired conclusion:

$$\nabla_\theta \log d^{\pi_\theta}(\bar{s}) = \lim_{T \to \infty} \sum_{j=0}^{T} \int q^{\pi_\theta}(s_{t+j} = s, a_{t+j} = a|s_{t+T+1} = \bar{s}) \nabla_\theta \log \pi_\theta(a|s) ds, a$$

$$= \lim_{T \to \infty} \sum_{j=0}^{T} \int q^{\pi_\theta}(s_t = s, a_t = a|s_{t+T+1-j} = \bar{s}) \nabla_\theta \log \pi_\theta(a|s) ds, a \tag{15}$$

$$= \lim_{T \to \infty} \int \sum_{j=0}^{T} q^{\pi_\theta}(s_t = s, a_t = a|s_{t+j+1} = \bar{s}) \nabla_\theta \log \pi_\theta(a|s) ds, a$$

# B  Hyperparameters

| General parameters | |
|---|---|
| Total iterations | $2e6$ |
| Batch size $B$ | 256 |
| $\gamma$ | 0.9 |
| Replay memory size | 50000 |
| $N_{\mathcal{B}}$ | 15000 |
| $N_\pi$ | 5000 |
| **$\mathcal{B}^s$ and $\mathcal{B}^a$** | |
| Stacked autoencoders | 2 |
| Hidden layers | 500, 500 |
| Optimizer | Adam |
| Learning rate | $2 \cdot 10^5$ |
| Burnin | 50000 iterations |
| L2-regularization | $10^{-2}$ |
| $min(\sigma_i)$ | 0.1 |
| Gradient clip, $L_2$ norm | 100 |
| **Policy $\pi_\theta$** | |
| Hidden layers | 300, 200 |
| Optimizer | Adam |
| Learning rate | $10^4$ |
| $\sigma$ bounds | (0.01, 0.1) |

(a) GPRIL parameters for clip insertion

| General parameters | |
|---|---|
| Total iterations | $1e4$ |
| #Processes | 16 |
| Batch size $B$ | $16 \cdot 256$ |
| Actor steps per iteration | 3 |
| Discriminator steps per iteration | 1 |
| $\gamma$ | 0.995 |
| **Actor** | |
| Hidden layers | 300, 200 |
| KL step size | 0.01 |
| $\sigma$ bounds | (0.01, 0.1) |
| **Discriminator** | |
| Hidden layers | 150, 100 |
| Optimizer | Adam |
| Learning rate | $10^4$ |
| Entropy regularization | 1 |
| Optimizer | Adam |
| **Critic** | |
| Hidden layers | 300, 200 |
| Optimizer | Adam |
| Learning rate | $5 \cdot 10^3$ |

(b) GAIL parameters for clip insertion

| General parameters | |
|---|---|
| Total iterations | $5e6$ |
| Batch size $B$ | 256 |
| Hidden layers | 300, 200 |
| $\sigma$ bounds | (0.01, 0.1) |
| Optimizer | Adam |
| Learning rate | $10^4$ |
| L2-regularization | $10^{-4}$ |

(c) BC parameters for clip insertion

| | Simulation | Real robot |
|---|---|---|
| **General parameters** | | |
| Batch size $B$ | 256 | |
| $\gamma$ | 0.9 | 0.7 |
| Replay memory size | 10000 | 50000 |
| $N_{\mathcal{B}}$ | 10000 | |
| $N_\pi$ | 1000 | 5000 |
| **$\mathcal{B}^s$ and $\mathcal{B}^a$** | | |
| Stacked autoencoders | 2 | |
| Hidden layers | 500, 500 | |
| Optimizer | Adam | |
| Learning rate | $10^{-5}$ | $3 \cdot 10^{-5}$ |
| Burnin | 0 | |
| L2-regularization | $10^{-2}$ | $10^{-3}$ |
| $min(\sigma_i)$ | 0.1 | 0.01 |
| Gradient clip ($L_2$) | 100 | |
| **Policy $\pi_\theta$** | | |
| Hidden layers | 300, 200 | |
| Optimizer | Adam | |
| Learning rate | $10^{-4}$ | |
| $\sigma$ bounds | (0.01, 0.1) | |

(d) GPRIL parameters for peg insertion

| General parameters | |
|---|---|
| #Processes | 16 |
| Batch size $B$ | $16 \cdot 256$ |
| Actor steps/iteration | 3 |
| Discriminator steps/iteration | 1 |
| $\gamma$ | 0.995 |
| **Actor** | |
| Hidden layers | 300, 200 |
| KL step size | 0.01 |
| $\sigma$ bounds | (0.01, 0.1) |
| **Discriminator** | |
| Hidden layers | 150, 100 |
| Optimizer | Adam |
| Learning rate | $10^{-4}$ |
| Entropy regularization | 1 |
| Optimizer | Adam |
| **Critic** | |
| Hidden layers | 300, 200 |
| Optimizer | Adam |
| Learning rate | $5 \cdot 10^{-3}$ |

(e) GAIL parameters for simulated peg insertion

## C    RELATION TO THE POLICY GRADIENT THEOREM

In this section, we outline the relation between the policy gradient theorem (Sutton et al., 1999) and the state-action-distribution gradient derived in section A and show equivalence of the discount factor used in reinforcement learning and the discount factor $\gamma$ introduced in this work. We first show that the state-action distribution gradient is equal to the policy gradient in the average reward case using a specific reward function. We then derive the $\gamma$-discounted approximation of the state-action-distribution gradient presented in section 3.1 from the policy gradient in the discounted reward framework using $\gamma$ as a discount factor. While this derivation is more cumbersome than the one presented in the body of the paper, it allows us to gain a better understanding of the meaning of $\gamma$ since discounting in the reinforcement learning setting is well understood. For notational simplicity, we assume that the state space $\mathcal{S}$ and action space $\mathcal{A}$ are countable. Note that this is a common assumption made in the field (Tsitsiklis and Van Roy, 1997) and considering that any state-action space is countable and finite when states and actions are represented using a finite number of bits makes it apparent that this assumption does not constitute a simplification in the practical sense.

In this work we propose to follow the gradient $\nabla_\theta \log \rho^{\pi_\theta}$ evaluated at demonstrated states and actions, consider thus the gradient evaluated at $\overline{s}, \overline{a}$:

$$
\begin{aligned}
\nabla_\theta \log \rho^{\pi_\theta}(\overline{s}, \overline{a}) &= \frac{1}{\rho_\theta^\pi(\overline{s}, \overline{a})} \nabla_\theta \sum_{s \in \mathcal{S}, a \in \mathcal{A}} \rho^{\pi_\theta}(s, a) \mathbf{1}(s = \overline{s}, a = \overline{a}) \\
&= \nabla_\theta \mathbb{E}_{s, a \sim \rho_\theta^\pi}[R(s, a)] =: \nabla_\theta J(\pi_\theta)
\end{aligned}
\tag{16}
$$

As we can see, the gradient is equivalent to the policy gradient using the reward function $R(s, a) := \frac{\mathbf{1}(s = \overline{s}, a = \overline{a})}{\rho_{\hat{\theta}}^\pi(\overline{s}, \overline{a})}$ in the average reward framework where $\hat{\theta}$ corresponds to the parameters of the policy at the current iteration. This reward function is not practical as it can be infinitely sparse and furthermore depends on the unknown stationary distribution. However, it allows us to derive the policy gradient using this reward function in the discounted reward setting which constitutes a well understood approximation of the average-reward scenario. Following the notation in Sutton et al. (1999), we have:

$$
\nabla_\theta J(\pi_\theta) = \mathbb{E}_{s_j, a_j \sim \rho_\theta^\pi}\left[\nabla_\theta \log \pi_\theta(a_j|s_j) \mathbb{E}_{s_{j+1}, a_{j+1}, \dots}\left[\sum_{t=0}^\infty \gamma^t R(s_{t+j}, a_{t+j})|\pi_\theta, s_j, a_j\right]\right]
\tag{17}
$$

We can now replace the expectation over the stationary distribution by an expectation over the path of the agent. This yields a double sum whose order of summation can be changed:

$$
\begin{aligned}
\nabla_\theta J(\pi_\theta) &= \lim_{T \to \infty} \frac{1}{T} \mathbb{E}\left[\sum_{j=0}^T \nabla_\theta \log \pi_\theta(a_j|s_j) \sum_{t=0}^T \gamma^t R(s_{t+j}, a_{t+j})|\pi_\theta\right] \\
&= \lim_{T \to \infty} \frac{1}{T} \mathbb{E}\left[\sum_{j=0}^T \sum_{t=j}^T \nabla_\theta \log \pi_\theta(a_j|s_j) \gamma^{t-j} R(s_t, a_t)|\pi_\theta\right] \\
&= \lim_{T \to \infty} \frac{1}{T} \mathbb{E}\left[\sum_{t=0}^T \sum_{j=0}^t \gamma^{t-j} \nabla_\theta \log \pi_\theta(a_j|s_j) R(s_t, a_t)|\pi_\theta\right]
\end{aligned}
\tag{18}
$$

After changing the order of summation we can replace the outer sum with the expectation over the stationary distributions and use the special nature of the chosen reward function to write the gradient as a conditional expectation:

$$
\begin{aligned}
\nabla_\theta J(\pi_\theta) &= \lim_{t \to \infty} \mathbb{E}\left[\sum_{j=0}^t \gamma^{t-j} \nabla_\theta \log \pi_\theta(a_j|s_j) R(s_t, a_t)|\pi_\theta\right] \\
&= \lim_{t \to \infty} \mathbb{E}\left[\sum_{j=0}^t \gamma^{t-j} \nabla_\theta \log \pi_\theta(a_j|s_j)|\pi_\theta, s_t = \overline{s}, a_t = \overline{a}\right]
\end{aligned}
\tag{19}
$$

Finally, we notice that this equation constitutes the discounted version of equation 15, thus we can immediately obtain our estimate of the state-action-distribution gradient:

$$\nabla_\theta J(\pi_\theta) = \frac{1}{1-\gamma} \mathbb{E}_{s,a \sim \mathcal{B}^{\pi_\theta}(\cdot,\cdot|\bar{s})} \left[ \nabla_\theta \log \pi_\theta(a|s) \right] + \nabla_\theta \log \pi_\theta(\bar{a}|\bar{s}) \approx \nabla_\theta \log \rho^{\pi_\theta}(\bar{s}, \bar{a}) \qquad (20)$$

While this derivation is less direct than the derivation used in section A, it draws a connection between the problem of matching state-action-distributions and the reinforcement learning problem with a reward that is positive for demonstrated state-action pairs and 0 otherwise. This indicates that the role of the discount factor is similar in both settings: Lower values of $\gamma$ trade-off accurate matching of distributions for lower variance and as expedience. Agents with low $\gamma$ will learn policies that recover and reach demonstrated states quicker over policies that are matching the experts state-action distribution more accurately long-term. Furthermore, the long history of successes in reinforcement learning validates the use of $\gamma$ as an approximation to the true objective which is often more accurately described by the average-reward objective.

