# OpenReview forum: "Generative predecessor models for sample-efficient imitation learning"
_ICLR.cc/2019/Conference_

### Official Review · AnonReviewer2 · 2018-11-01
**Compelling, sample efficient approach to imitation learning using learned dynamics models. Experiments could be extended.**

**Rating:** 7
**Confidence:** 4

**Review:**

The submission builds up on recent advances in neural density estimation to develop a new algorithm for imitation learning based on a probabilistic model for predecessor state dynamics. In particular, the method trains masked autoregressive flows as a probabilistic model for state action pairs conditioned on future states. This model is used to estimate the gradient of the stationary distribution of a policies visited states. Finally, the proposed objective uses this estimate and the gradient of the log likelihood of expert actions under the policy to maximise the similarity of the expert’s and agent’s stationary state-action distributions.

The proposed method outperforms existing imitation learning approach (GAIL & BC) on 2 simulation-based manipulation tasks. It performs particularly well in terms of sample efficiency.
The magnitude of difference between the sample efficiencies of GAIL and the proposed approach seems quite surprising and it would be beneficial if the authors could explicitly state if the measured number of samples include the ones used for training of the probabilistic model as well as the policy (apologies if I have missed a section fulfilling this purpose).

While the improvements on the presented experiments are clear, the experimental section represents a small shortcoming of the submitted paper. The 2 experiments (clip and peg insertion) are quite similar in type and to not take into account other common domains e.g. locomotion tasks from the original GAIL paper. Furthermore, an additional comparison to SAIL would be recommended since the approaches are closely related as the authors acknowledge. The provided comparison with different types of available expert data is quite interesting and could possibly be extended to test other state-of-the-art methods (action-free versions of GAIL, AIRL,etc.).

Nonetheless, the paper overall presents a strong submission based on novelty & relevance of the proposed method and is recommended for publication.

Minor issues:
- Related work: improve transitions between the section about trajectory tracking and BC.
- Ablation studies with less flexible probabilistic models would strengthen the experiment section further.
- Add derivation from Eq. 3 to 4 and 5 to appendix to render the paper more self-contained and easier to access.
- A release of the code base would further strengthen the contributions of the submission.

General recommendation:
- The authors are encouraged to further investigate off-policy corrections for improved convergence.

---

> ### Author Response · Authors · 2018-11-09
> **Changes made to the manuscript & addressing questions raised.**
>
> Thank you for your review. We would like to address some of the questions and points raised in your review:
>
> + Regarding sample efficiency, our algorithm only uses artificial samples and expert samples when updating the policy. The reported number of environment samples are the samples used to train the generative model. Note that while it is efficient compared to other algorithms of this type, the data efficiency is not extraordinary to train a network of this size on a problem of these dimensions. This indicates that the samples generated from the model are likely to be useful even as the policy changes.
> + Regarding the chosen domains, we believe that they highlight a type of problem that is difficult for approaches based on adversarial updates (due to sample efficiency as well as the ability to control the state in its entirety) while also being representative for a class of problems one might encounter in practice. While we believe our approach to be widely applicable and would like to see it applied in other domains, existing approaches are already able to achieve very high scores on domains such as the mujoco walkers.
> + In comparison to SAIL, we aim to address scalability w.r.t. the number of parameters of the policy. We found that larger policies are able to achieve more accurate results on our domains, yet policies of this size are out of reach for SAIL which has to predict the gradient for each parameter of the network.
> + We added additional steps to Appendix A to make the derivation more self-contained and changed the wording in the related works section based on your suggestion. Unfortunately, we are currently not able to release the code.

---

> > ### Comment · AnonReviewer2 · 2018-11-21
> > **Response**
> >
> > Thanks for the feedback and clarification regarding the use of no environment-interaction samples for training the policy.
> > As emphasised in the original review, the experimental section remains a shortcoming and to demonstrate versatility other tasks would have to be included. If the approach is robust enough this should be fairly little effort and the code for e.g. the original GAIL environments is openly available (https://github.com/openai/imitation).
> > By missing the comparison to some of the most closest (SAIL) or current state-of-the-art approaches (AIRL, etc) the authors further weaken the experimental section in an otherwise stronger paper.
> > Based on the these shortcomings the paper cannot fully evaluate the proposed approach.

---

### Official Review · AnonReviewer1 · 2018-11-02
**Good Results But Relevant Literature is missing.**

**Rating:** 5
**Confidence:** 5

**Review:**

The paper proposes to use predecessor models for imitation learning to overcome the issue of only observing expert samples during imitation from expert trajectories.

The paper is very well written. But the proposed method is really not novel. The idea of using predecessor models have already been explored in multiple places [1], [2] (but not in imitation learning scenario!). Hence, the novelty comes from using the predecessor models for imitation learning. The introduction of the paper should mention this  to reflect the contribution.

[1] Recall Traces: Efficient Backtracking models for efficient RL https://arxiv.org/abs/1804.00379
[2] Organizing Experience: A Deeper Look at Replay Mechanisms for Sample-based Planning in Continuous State Domains
https://arxiv.org/abs/1806.04624

Both of these papers should be cited and discussed.

Results: The proposed method outperforms GAIL and behaviour cloning in terms of sample efficiency   on simulation-based manipulation tasks.

Regarding experiments, I would like to see certain baselines.

- What happens when you predict sequentially using predecessor models ? I understand that the sequential generation is prone to accumulating errors, but as [1] points out, using predecessor models you can sample from many states on the expert trajectory. And Hence possible to get good learning signal even while sampling shorter trajectories using predecessor models.

- Comparison with Dyna based methods. For this baseline, authors would learn a forward model. And then sample from the forward model, and use the samples from the forward model for imitation learning.

---

> ### Author Response · Authors · 2018-11-09
> **Specifying the scope of our work and additional citations.**
>
> Thank you for your review. Predecessor models do indeed have a long history in reinforcement learning and recent work explores the use of deep networks in this context. While our algorithm followed directly from the derivation of the state-distribution gradient in section 3.1, we can see that a comparison to the aforementioned works might be useful to the reader and have added this in section 1. We hope that this addition will adequately specify the scope of our work. In particular, we claim the following two contributions:
>
> 1. Derivation of the state-distribution gradient based on samples from a predecessor model. While such models have been used in the past, to the best of our knowledge this connection has not been pointed out before. Instead, most work focuses on the use of predecessor models as a more efficient order of bellman backups while the recent Recall Traces uses a justification based on a variational lower bound. We believe that our derivation provides further justification to the approach used in Recall Traces and may furthermore help to guide design decisions when developing such algorithms in the field of reinforcement learning.
> 2. Development of a novel, state-of-the-art imitation learning algorithm. To the best of our knowledge, the use of predecessor models to achieve state-action distribution matching in imitation learning is novel. We believe that predecessor models are a natural fit for imitation learning as, unlike in reinforcement learning, future observations and their accordance with demonstrations are very difficult to evaluate. We demonstrate the effectiveness of such models on traditionally difficult real world imitation learning problems in our evaluation.
>
> Regarding our choice of multi-step models and comparison to one-step models of either direction, we note that in the general case, the error in naive one-step models grows exponentially (Venkatraman et al., 2015) thus requiring careful design of such models. Recent work such as Ha and Schmidhuber, 2018 and Gregor and Besse, 2018 achieves impressive predictions on sequential rollouts indicating that it is very likely that a one-step model can be applied in our setting as well. However, these works require a significant effort on the modelling side and we thus decided to side-step the issue by modelling the desired distribution directly. As the contact dynamics in our domain can be complex, we believe that the effort required in our domain would have been significant as well. We note that the main contribution of this paper is the use of samples of a predecessor model to match state-action distributions in a principled way while the choice of model is a design choice that was made to avoid increasing complexity.

---

> > ### Comment · AnonReviewer1 · 2018-11-19
> > **Reply**
> >
> > Sorry for late reply.
> >
> > "the error in naive one-step models grows exponentially "
> >
> > Learning any kind of model (whether forward model or predecessor model) for more than (lets say 10 steps) is challenging task. This paper needs to argue (by showing experiments empirically) that using predecessor models is better as compared to these 2 baselines.
> >
> > - When the agent learns a jumpy forward model. (Also used in learning to query paper [1]) as compared to jumpy predecessor model.
> > - Unrolling the predecessor model step by step.
> > - Lets say you have a trajectory of length 1000, then one can take every 5th state on this expert trajectory, and generate traces (according to the learned forward model) and use these generated traces for imitation learning. As the authors pointed out that unrolling the forward model (or predecessor model ) is prone to compounding errors, but one can easily make predictions for 5/10 steps. And since you have an expert trajectory, you can use every kth state as an input to forward model for generating traces (here k==5/10). This wont be prone to "compounding errors".
> >
> > Unless until authors compare to these baselines, I dont think the contribution is justified.
> >
> > [1] https://arxiv.org/abs/1802.03006

---

> > > ### Author Response · Authors · 2018-11-23
> > > **Regarding the suggested baselines**
> > >
> > > Thank you for elaborating on your review. The main contributions of our paper are 1. a state-of-the-art imitation learning algorithm as outlined in Algorithm 1 and 2. an algorithmic framework using predecessor models to estimate the state-distribution-gradient. To the best of our knowledge and including the new references suggested in the review, these contributions are significantly different from prior work.
> > > We agree that a comparison to unrolled forward or backwards models is very interesting.  However, there are numerous ways to approach this, as the reviewer points out, and providing one that isn’t a straw-man is non-trivial for reasons discussed below.  We view our work instead as providing the theoretical foundation for an alternative approach that does not rely on unrolling, and focused our efforts on demonstrating viability on a real-world task setting.  In our view, novel theoretical work, real-world experiments, and comparison to potential alternatives for model components that lack established baselines is too dense for a single paper.  We note that we did perform evaluation against GAIL, which is an established baseline for our task setting.
> > >
> > > 1. Our contribution is a novel imitation learning approach that can work with either, a jumpy or a single-step predecessor model. The choice of model does not matter to the applicability of our algorithm. While we believe that a jumpy model will work better, our specific contribution would remain the same if we had found that a single-step approach works just as well.
> > > 2. Unlike in reinforcement learning, where a forward-model may trivially be incorporated, the use of a forward-model is non-trivial in imitation learning. We provide some analysis relating the time-reversed approach to the forward view in Appendix C. The reviewer suggests a Dyna-like approach, perhaps referring to a combination of Dyna and GAIL and perhaps incorporating the suggested work by Buesing et al.. We are not aware of successful application of such a combination and it is unclear whether it would be stable. However, work submitted to this conference (https://openreview.net/forum?id=Hk4fpoA5Km) aims to evaluate a very similar proposition by using GAIL with a replay buffer. We are excited to see other investigations into sample efficient imitation learning and believe that adversarial objectives will continue to have their place in the field. However, we would like to point out that while the work in question does significantly increase sample-efficiency, it does not reach the level of real-world applicability and is furthermore clearly concurrent.
> > >
> > > We compare against a widely accepted state-of-the-art imitation learning algorithm as a baseline and demonstrate that the approach works on a real robot platform. We believe that those experiments validate our claims but hope to join other researchers future work on the development and evaluation of optimized, reverse-time models as well as novel imitation-learning algorithms utilizing jumpy forward models.

---

> > > > ### Comment · AnonReviewer1 · 2018-11-23
> > > > **Thanks!**
> > > >
> > > > Thanks for your time in engaging in discussions with the reviewer. The reviewer appreciates it.
> > > > As of now, I dont think, I'm convinced.
> > > >
> > > > " a state-of-the-art imitation learning algorithm"
> > > >
> > > > It could be. But by only comparing the proposed approach on similar envs, I'm not sure it makes sense to call your proposed method (which already been used for RL) a "state of the art imitation learning algorithm"
> > > >
> > > > "We agree that a comparison to unrolled forward or backwards models is very interesting."
> > > >
> > > > This SHOULD be the baselines. Until unless, authors compare to these baselines, the reviewer does not know how novel the proposed method is.....
> > > >
> > > > " In our view, novel theoretical work, real-world experiments, and comparison to potential alternatives for model components that lack established baselines is too dense for a single paper."
> > > >
> > > > This is not DENSE. These SHOULD be the baselines. Asking the right baselines, is my job. I'm not criticizing the method. I have already pointed out that I *REALLY* enjoyed  reading the paper, and paper is well written.    I'm not really sure, why the authors dont want to do this.  Taking your code, and comparing to the unrolled backtracking model should be minimal change. And comparing to DYNA is a natural baseline.
> > > >
> > > > Even just compare to BC (behaviour cloning).
> > > >
> > > > Pseudo Code (Baseline 1)
> > > >
> > > > Take the expert trajectories (s, a, s_t+1)
> > > > Learning a forward model.
> > > > Generate traces from every kth state from the expert trajectories.  (where k==1/2/5/10)
> > > >
> > > > Pseuco Code (Baseline 2)
> > > >
> > > > Unrolled backward model (or predecessor model)
> > > >
> > > > (Edit:1) P.S - It may be possible that the reviewer is missing something, because of which its technically not possible to compare to reviewers, if thats the case, I would appreciate if the authors could point it out.
> > > >
> > > > Thanks!

---

> > > > > ### Author Response · Authors · 2018-11-23
> > > > > **Regarding forward models and BC**
> > > > >
> > > > > Thank you. At this point, we would like to give a quick reply regarding a single aspect of your response and request further clarification.
> > > > >
> > > > > Specifically, our reply is regarding a comparison against behavioral cloning, which we presume is meant to be in combination with a forward model:
> > > > > + In a forward model, the sampled actions are usually generated by the current policy. Note that E_π[∇log π(a|s)|s]=0 for all states. Therefore, performing behavioral cloning on rollouts of a well-trained forward model is not changing the policy in a meaningful way.
> > > > > + We have experimented with a variant of this where the states are taken from expert trajectories while the actions are generated using an action model that is conditioned on the target state, similar to B(a|s,s’) in our work. It is possible that this approach may work in some domains but we found that it leads to a self-reinforcing loop in our experiments. We believe this to be due to the following reason: If the robot is drifting to the right of the expert trajectory at the beginning of training, the conditioned action model will be trained with a strong prior that emphasizes actions that cause the drift. In our approach, we find that this is counteracted by our model generating states that are far to the right of the expert-trajectory, providing samples in the part of the state-space where the conditioned action model is trained as well as a larger discrepancy to the target which forces the action model to correct its course. If we instead use expert states, this effect is missing and the effect obtained by conditioning on the target is much weaker. Thus, we observe the prior taking over and the drift to be reinforced.
> > > > >
> > > > > Please let us know if either of these variations corresponds to the suggested baseline.

---

> > > > > > ### Comment · AnonReviewer1 · 2018-11-23
> > > > > > **Quick reply**
> > > > > >
> > > > > > "In a forward model, the sampled actions are usually generated by the current policy. Note that E_π[∇log π(a|s)|s]=0 for all states. Therefore, performing behavioural cloning on rollouts of a well-trained forward model is not changing the policy in a meaningful way. "
> > > > > >
> > > > > > Right. I was thinking that you can learn a forward model from the expert trajectories. And then using the forward model you can sample traces from every kth state (k ==5/10),
> > > > > > and using the sampled traces, you can use those traces for imitation learning (i.e supervised learning).
> > > > > >
> > > > > > I am just about to board a flight in next 1 min, so would reply to the other part in few hours. Sorry about that.

---

> > > > > > > ### Author Response · Authors · 2018-11-23
> > > > > > > **Forward models on expert data**
> > > > > > >
> > > > > > > I believe I understand. However, in order to generate training samples for supervised learning, the forward model would have to learn to predict actions as well as states.
> > > > > > > If the model is trained based on expert data alone, it would therefore have to learn to predict actions for each state based on the demonstrations. This is essentially the behavioral cloning problem which we have found to either overfit given the amount of data provided or to learn a very weak policy due to the amount of regularization required. Methods such as GAIL or GPRIL are able to combat overfitting by learning more about the environment using their own, self-supervised, experience. If the action prediction part of the model is trained using samples generated from the current policy, however, the expectation of the supervised learning gradient using generated training samples will be 0. Please also note that in a large number of our experiments, including the experiment on the physical robot, we are not using demonstrated actions at all and are training the agent from demonstrated states alone. In such scenarios, a method that trains a model based on expert data alone would not be able to predict actions at all.
> > > > > > >
> > > > > > > Please let me know if my assumptions are correct.

---

> > > > > > > > ### Comment · AnonReviewer1 · 2018-11-25
> > > > > > > > **thanks for your reply**
> > > > > > > >
> > > > > > > > "the forward model would have to learn to predict actions as well as states."
> > > > > > > >
> > > > > > > > You would be learning a policy as well...
> > > > > > > >
> > > > > > > > "If the model is trained based on expert data alone, it would therefore have to learn to predict actions for each state based on the demonstrations. This is essentially the behavioural cloning problem which we have found to either overfit given the amount of data provided or to learn a very weak policy due to the amount of regularization required. "
> > > > > > > >
> > > > > > > > I'm not sure, I understand the argument. You would learn a model, and then same traces from the model, and use the generated traces for imitation learning.
> > > > > > > >
> > > > > > > > " Please also note that in a large number of our experiments, including the experiment on the physical robot, we are not using demonstrated actions at all and are training the agent from demonstrated states alone. In such scenarios, a method that trains a model based on expert data alone would not be able to predict actions at all."
> > > > > > > >
> > > > > > > > Yes. I agree with this point.
> > > > > > > >
> > > > > > > > Still, it remains, that the authors have not yet compared the proposed method with a simple baseline of unrolling the backwards model, and generating traces from different starting positions.

---

> > > > > > > > > ### Author Response · Authors · 2018-11-25
> > > > > > > > > **learning a policy as well**
> > > > > > > > >
> > > > > > > > > To clarify based on my understanding:
> > > > > > > > > * In the backward view, we train a model to generate trajectories that end at a demonstrated state and thus train the agent to recover. The model is factorized split into B(s|s') and B(a|s,s'), the latter corresponds to a policy  that has been given additional information which enables learning.
> > > > > > > > > * For forwards rollouts, we would generate trajectories that start at a demonstrated state. After that, it would simply follow the policy that the forward model has learned. If this policy is trained from expert demonstrations, the model therefore has to solve the behavioral cloning problem. We already compare against BC and show that it doesn't learn a good policy.
> > > > > > > > >
> > > > > > > > > Please let me know if this scenario fully describes the suggested baseline.

---

### Official Review · AnonReviewer3 · 2018-11-05
**interesting idea, but some issues need to be clarified.**

**Rating:** 6
**Confidence:** 3

**Review:**

This paper studies the problem of matching the state-action distributions of agent and expert demonstrations. In order to address this problem, the authors consider a likelihood treatment comprising a conditional probability (which is estimated from demonstrations) and a state distribution (which is estimated from sampling approximations).

The authors provide a descent result (i.e., equ. (7)) to estimate the gradient of the logarithmic state distribution. One problem is that it is unclear how the discount factor $\gamma$ influence this result?

In addition, in (12), two scaling factors are used, so how to balance these weights?

Specifically, in (11), it seems the authors are considering the stationary joint state-action distribution, which is different from the state-action distribution generated by the agent on-line, it is suggested to clarify this issue.

---

> ### Author Response · Authors · 2018-11-09
> **Additional discussion added concerning the effect of approximations made in the paper.**
>
> Thank you for your review. The questions you raise about the effect of the approximations made in our derivation are valid and we have added additional discussion to the paper that we hope will answer these questions satisfactorily:
>
> + The discount factor γ is not only similar to the discount factor used in reinforcement learning but can be seen as identical. This was not immediately apparent in the original submission and we have added Appendix C to the manuscript to explore the connection between our gradient estimate and policy gradients. As a result, we can draw on the understanding of the discount factor in reinforcement learning to gain insight into the behavior of γ and conclude that first, the discount factor introduces a trade-off where an agent with lower γ prefers to reach demonstrated states more quickly while agents with higher γ will aim to reproduce the state-action distribution more closely in the long-term. Second, as γ approaches 1, the variance grows. However, in reinforcement learning, it has been empirically shown that lower discount factors can be an accurate, low-variance approximation even when the true objective is more accurately described by the average reward objective (γ -> 1). The alternative derivation introduced in Appendix C thus indicates that lower values of γ are likely to be reasonable approximations in the imitation learning setting as well.
> + With regards to stationarity, under the usual ergodicity assumptions the expected distribution of state-action pairs the agent will observe and the stationary distribution should be identical in the infinite horizon case (using the modified MDP with terminal states being treated as transitions to initial states as discussed in section 2.1). In general, matching the joint stationary distribution to the empirical distribution of the expert implies a form of loop in the agents behavior which may be as simple as restarting after reaching a terminal state. This is the case in many practical scenarios as well as our experiments. While handling the finite horizon case explicitly might also be interesting, we are not considering it for the purposes of this work.
> + The scaling factors β were added to provide more freedom to tune the behavior of the learning algorithm but we agree that additional discussion would be useful and have added it to section 3.3. In particular, the factors are the result of dropping the factor of 1/(1-γ) in equation 7, this indicates that a sensible starting point would be β_π=(1-γ)β_d. However, we did not find this to be optimal in all cases. In particular, if behavioral cloning is likely to overfit strongly, lower values of β_π may be adequate while in cases where exploring to learn the generative models is more difficult, higher values of β_π may provide more guidance.
>
>
> We hope that we answered your questions to your satisfaction, please let us know if you have further questions or concerns that you would like us to address.

---

### Meta-Review · Area_Chair1 · 2018-12-15

**Confidence:** 4
**Recommendation:** Accept (Poster)

**Metareview:**

This paper proposes to estimate the predecessor state dynamics for more sample-efficient imitation learning. While backward models have been used in the past in reinforcement learning, the application to imitation learning has not been previously studied. The paper is well-written and the results are good, showing clear improvements over GAIL in the presented experiments. The primary weakness of the paper is the lack of comparisons to the baselines suggested by reviewer 1 (a jumpy forward model and a single step predecessor model) to fully evaluate the contribution, and to SAIL and AIRL. Despite these weaknesses, the paper slightly exceeds the bar for acceptance at ICLR.
The authors are strongly encouraged to include these comparisons in the final version.